# Beyond Topological Persistence: Starting from Networks

**Mattia G. Bergomi** [1], **Massimo Ferri** [2,*], **Pietro Vertechi** [1] and **Lorenzo Zuffi** [2]

1 Veos Digital, 20124 Milan, Italy; mattia.bergomi@veos.digital (M.G.B.); pietro.vertechi@veos.digital (P.V.)
2 Advanced Research Center on Electronic Systems "E. De Castro", Department of Mathematics, Università di Bologna, 40126 Bologna, Italy; lorenzo.zuffi@studio.unibo.it
* Correspondence: massimo.ferri@unibo.it

**Abstract:** Persistent homology enables fast and computable comparison of topological objects. We give some instances of a recent extension of the theory of persistence, guaranteeing robustness and computability for relevant data types, like simple graphs and digraphs. We focus on categorical persistence functions that allow us to study in full generality strong kinds of connectedness—clique communities, *k*-vertex, and *k*-edge connectedness—directly on simple graphs and strong connectedness in digraphs.

**Keywords:** categorical persistence function; connectedness; persistence diagram; poset; graph; digraph

**MSC:** 55N31; 62R40; 68R10; 05C10; 18C99

## 1. Introduction

Persistent homology allows for swift and robust comparison of topological objects. However, raw data are rarely endowed with a topological structure. Persistent homology and topological persistence are by their nature bound to topological spaces and simplicial complexes, so that, in persistent homology applications, data are mapped to topological spaces or simplicial complexes through auxiliary constructions (e.g., [1–5]). Although these constructions have been employed successfully in several domains (e.g., [6–10]), they unavoidably transform the information carried by the original data set. Moreover, in some cases it is not possible to create such constructions directly, due to the lack of fundamental properties. For instance, the blocks of a filtered graph do not form a simplicial complex because blocks lack the hereditariness that is required by simplicial constructions. In particular, directed graphs generally present additional difficulties, although [11] overcomes them elegantly, via a poset-based technique (an idea that we borrow and further develop in this work). Thus, it would be advantageous to be able to use the tools of persistence also if one is interested in graph-theoretical structures, which do not enjoy the hereditary property necessary for building simplicial complexes. This paper aims to promote this wider use of persistence in graphs.

Rank-based persistence [12] extends topological persistence to arbitrary categories from an axiomatic perspective. In summary, the theory developed in [12] allows one to compute the persistence of objects in arbitrary source categories (rather than topological spaces) and consider any regular category as the target category, whereas classical persistence is limited to vector spaces and sets.

We are well aware of greatly developed extensions of persistence beyond the topological or simplicial categories. [13–18] widely extend the range of target categories. In particular, by giving the possibility to use integer coefficients for homology (thus admitting torsion), these extensions can offer new perspectives to applications and theoretical developments. We are interested in extending the domain category, in the spirit of [19]. We stick to the approach developed by [12], which is axiomatic, consequently providing swift

conditions to verify that a function is a persistence function. See Section 2, in particular, Definition 1.

Our main aim is to build on the aforementioned categorical generalization to allow for more direct analysis of significant data types such as simple graphs and guarantee the properties of stability and universality of the classical persistence framework. We hope that the axiomatic foundation mentioned above will yield a tool agile enough to enable the usage of persistence in applications so far not liable to immediate, direct topological constructions (although a more elaborate one exists; see Remark 1).

With this aim in mind, we introduce the definition of *weakly directed properties* as a way to easily build categorical persistence functions that describe graph-theoretical concepts of connectivity, e.g., clique communities, *k*-vertex and *k*-edge connectedness in graphs, and strong connectedness in digraphs. In more detail, in Section 2, we define monic persistence functions and show how the persistence diagrams associated with such functions can be described as multisets of points and half-lines, as in the classical framework. Furthermore, we define the natural pseudodistance in this general context and list the assumptions necessary to obtain tame filters. In Section 2.2, we introduce our framework in the category of weakly directed posets. We define stable monic persistence functions in this category and prove stability and universality. This construction allows us to define *weakly directed properties* and describe the associated persistence functions.

In Section 3 (possibly the one of main interest for the application-oriented reader), we show that clique communities, *k*-vertex, and *k*-edge connectedness in graphs, and strong connectedness in digraphs are weakly directed properties, and thus yield persistence functions, and persistence diagrams. Their stability and universality are discussed.

All constructions built through the proposed generalized persistence are discussed on the weighted graph depicted in Figure 1. For completeness, in the same figure we compute persistent Betti numbers of the example graph, seen as a simplicial complex.

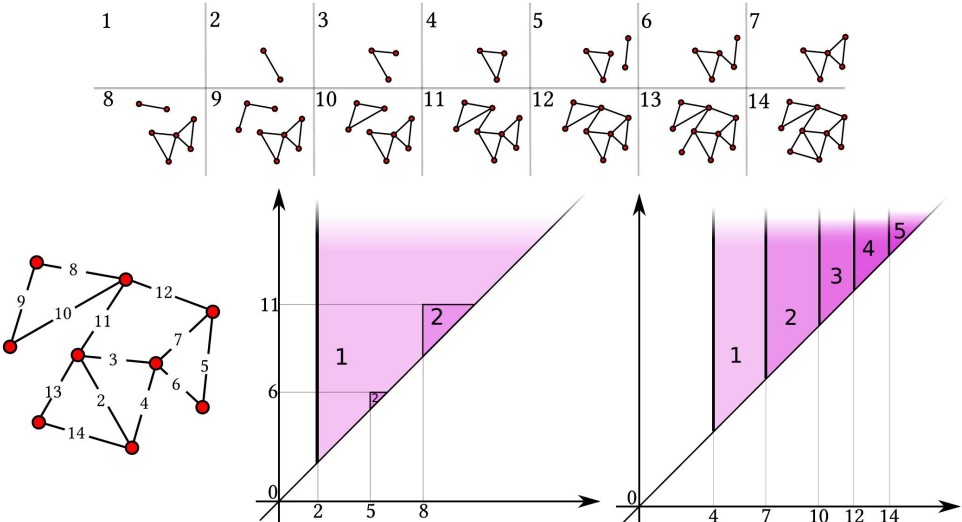

**Figure 1.** A weighted graph (**left**), the corresponding filtration (**above**), and its persistent Betti numbers functions of degree 0 (**middle**) and 1 (**right**).

## 2. Persistence via the Poset of Subobjects

We give concrete applications of the framework developed in [12], which defines the notion of *categorical persistence function* ([12], Def. 3.2) in arbitrary categories. Unlike [12], here we restrict ourselves to filtrations, rather than arbitrary $(\mathbb{R}, \leq)$-indexed diagrams. In other words, given a category **C**, we will consider categorical persistence functions in $\mathbf{C}_m$, the subcategory of **C** where the only allowed morphisms are monomorphisms.

**Definition 1** ([12], Def. 3.2). *A* persistence function *is a categorical persistence function on the category* $(\mathbb{R}, \leq)$. *It is a correspondence that maps each pair of real numbers* $u \leq v$, *to an integer* $p(u,v)$ *such that, given* $u_1 \leq u_2 \leq v_1 \leq v_2$, *the following inequalities hold.*

1. $p(u_1, v_1) \leq p(u_2, v_1)$ *and* $p(u_2, v_2) \leq p(u_2, v_1)$, *that is to say p is non-decreasing in the first argument, and non-increasing in the second.*
2. $p(u_2, v_1) - p(u_1, v_1) \leq p(u_2, v_2) - p(u_1, v_2)$.

**Definition 2.** *Let* **C** *be an arbitrary category. A* monic persistence function *on* **C** *is a categorical persistence function on* $\mathbf{C}_m$. *It maps each inclusion* $u \hookrightarrow v$ *to an integer* $p(u,v)$, *such that, given* $u_1 \hookrightarrow u_2 \hookrightarrow v_1 \hookrightarrow v_2$, *the following inequalities hold.*

1. $p(u_1, v_1) \leq p(u_2, v_1)$ *and* $p(u_2, v_2) \leq p(u_2, v_1)$.
2. $p(u_2, v_1) - p(u_1, v_1) \leq p(u_2, v_2) - p(u_1, v_2)$.

A filtration $F$ in **C** can naturally be seen as a functor from $(\mathbb{R}, \leq)$ to $\mathbf{C}_m$. Therefore, by functoriality, a monic persistence function $p$ on **C** and a filtration $F$ in **C** induce a persistence function $p_F$. In turn, $p_F$ induces a persistence diagram $\mathcal{D}F$ as follows. By convention, in the following definition we consider $p_F(u,v) = \min_{x,y} p_F(x,y)$ whenever either $u$ or $v$ is not finite.

**Definition 3** ([12], Def. 3.13). *Given* $u, v \in \mathbb{R} \cup \{-\infty, +\infty\}$, $u < v$, *we define the multiplicity of* $(u,v)$, *denoted* $\mu(u,v)$, *as the minimum of the following expression over* $I_u, I_v$ *disjoint connected neighborhoods of u and v respectively:*

$$p_F(\sup(I_u), \inf(I_v)) - p_F(\inf(I_u), \inf(I_v)) - p_F(\sup(I_u), \sup(I_v)) + p_F(\inf(I_u), \sup(I_v))$$

*Whenever* $\mu(u,v) > 0$, *we say* $(u,v)$ *is a cornerpoint. The persistence diagram* $\mathcal{D}F$ *associated with* $p_F$ *is the multiset of its corner points, each with its multiplicity, along with all diagonal points* $(u,u)$, $u \in \mathbb{R}$, *with infinite multiplicity. A cornerpoint at infinity* $(u, +\infty)$ *is often (and here) represented as the straight half-line* $x = u, y \geq x$.

It is then possible to extend to this setting the classical notion of bottleneck (formerly matching) distance.

**Definition 4** ([12], Def. 3.24). *Let p be a monic persistence function and* $F_1, F_2$ *be filtrations in Ct; let* $\mathcal{D}F_1, \mathcal{D}F_2$ *be the respective persistence diagrams. The* bottleneck distance *between* $\mathcal{D}F_1$ *and* $\mathcal{D}F_2$ *is defined as*

$$d(\mathcal{D}F_1, \mathcal{D}F_2) = \inf_{\beta \in \mathcal{B}} \sup_{(u,v) \in \mathcal{D}F_1} \left\| (u,v) - \beta\big((u,v)\big) \right\|_\infty$$

*where* $\mathcal{B}$ *be the collection of all bijections from* $\mathcal{D}F_1$ *to* $\mathcal{D}F_2$.

Importantly, the appearance of persistence diagrams as the familiar multiset of points and half-line segments is guaranteed by the following proposition, where $p_F, \mu, \mathcal{D}F$ are as above and $\Delta^* = \{(x,y) \in \mathbb{R} \times (\mathbb{R} \cup \{+\infty\}) \mid x < y\}$.

**Proposition 1.** *For* $p_F$ *and* $\mathcal{D}F$ *we have*

$$p_F(\beta, \gamma) = \sum_{\substack{(u,v) \in \Delta^*, \\ u < \beta, \, v > \gamma}} \mu(u,v)$$

*for every* $(\beta, \gamma) \in \Delta^*$ *which is no discontinuity point of* $p_F$.

**Proof.** By applying ([12], Prop. 3.17) with $\alpha < \bar{a}$ and $\delta = +\infty$. $\square$

This result implies that the discontinuity sets of $p_F$ are either vertical or horizontal (possibly unbounded) segments with end-points in the corner points. This means that persistence functions have the appearance of superimposed triangles, typical of persistent Betti number functions.

**Natural pseudodistance.** It is possible to define the natural pseudodistance on $\mathbf{C}_m$. In the following, we will adopt some finiteness assumptions, to ensure stability of all persistence functions we will consider. Note that, whenever we refer to categories such as **Set**, **Poset**, **Graph**, **Digraph**, we always refer to the finite version—finite sets, finite posets, and finite simple graphs and digraphs.

**Finiteness assumptions.** From now on, we assume that every object in **C** has only a finite number of distinct subobjects (to ensure tameness in all constructions). Furthermore, we will only consider filtrations $F$ that admit a colimit $F(\infty)$ in $\mathbf{C}_m$. As every object has only a finite number of distinct subobjects, this means that $F(x \leq x')$ is an isomorphism for sufficiently large $x, x'$. This will allow us to define the *natural pseudodistance* [13,20,21].

**Definition 5.** *Let $F_1, F_2$ be two filtrations in **C**. Let $\mathcal{H}$ be the (possibly empty) set of isomorphisms between $F_1(\infty)$ and $F_2(\infty)$. Given an isomorphism $\mathcal{H} \ni \phi\colon F_1(\infty) \to F_2(\infty)$, we can consider the set*

$$L_\phi = \{\, h \in \mathbb{R}_{\geq 0} \mid \text{for all } x \in \mathbb{R}, \ F_2(x - h) \subseteq \phi(F_1(x)) \subseteq F_2(x + h) \,\},$$

*where the inclusion is among subobjects of $F_2(\infty)$. The* natural pseudodistance *between $F_1$ and $F_2$ is*

$$\delta(F_1, F_2) = \inf \bigcup_{\phi \in \mathcal{H}} L_\phi.$$

The natural pseudodistance is equal to the interleaving distance, when considering $F_1, F_2$ as $(\mathbb{R}, \leq)$-indexed diagrams in $\mathbf{C}_m$. By the universal property of colimits, a strong interleaving induces an isomorphism $F_1(\infty) \simeq F_2(\infty)$, which behaves correctly on sublevels. Conversely, applying $\phi$ and its inverse on sublevels induces a strong interleaving.

**Stability and universality** For applications, a wished-for quality is stability. A categorical persistence function $p$ on **C** is said to be *stable* if, given filtrations $F_1, F_2$ in **C**, for the induced $p_{F_1}, p_{F_2}$ and the corresponding persistence diagrams $DF_1, DF_2$ the inequality

$$d(\mathcal{D}F_1, \mathcal{D}F_2) \leq \delta(F_1, F_2)$$

holds. Moreover, the bottleneck distance is said to be *universal* with respect to $p$, if it yields the best possible lower bound for the natural pseudodistance, among the possible distances between $\mathcal{D}F_1$ and $\mathcal{D}F_2$, for any $F_1, F_2$. See Proposition 5; see [13] for a general discussion on universality.

*2.1. Preliminaries on Posets*

The first category we consider is **WDPoset**, the category of finite *weakly directed* posets.

**Definition 6.** *A poset $P$ is* weakly directed *if, whenever $a, b \in P$ have a lower bound, they also have an upper bound.*

Stong [22] discusses the analogy between finite posets and finite $T0$ topological spaces. The two categories are equivalent, and, therefore, it is possible to consider the homotopy type of a finite poset. In particular, Stong shows a procedure to determine whether two posets have the same homotopy type. For the sake of self-containment, we report here a description of the procedure from [23].

Let $P$ be a poset. An element $p \in P$ is *upbeat* (resp. *downbeat*) if the set of all element strictly greater (resp. lower) than $p$ has a minimum (resp. maximum). The insertion or deletion of an upbeat or downbeat point does not change the strong homotopy type of $P$. The core of $P$, denoted core($P$), is a deformation retract of $P$ that is minimal; i.e., it contains

neither upbeat nor downbeat elements. One can always reach core($P$) by successively deleting beat points from $P$.

**Theorem 1** ([22], Thm. 4). *Two finite posets are strongly homotopy equivalent if and only if they have isomorphic cores.*

We are now ready to show that there is a canonical homotopy equivalence between weakly directed finite posets and finite sets.

**Proposition 2.** *Let* Free : **Set** → **WDPoset** *be the free poset functor, i.e., the functor that associates to a finite set S the weakly directed poset* $(S, =)$. *Free admits a left adjoint* M. *Furthermore, let*

$$\epsilon : \mathrm{M} \circ \mathrm{Free} \to \mathrm{Id}_{\mathbf{Set}} \qquad and \qquad \eta : \mathrm{Id}_{\mathbf{WDPoset}} \to \mathrm{Free} \circ \mathrm{M}$$

*be the natural transformations associated to the adjunction.* $\epsilon$ *is a natural isomorphism, whereas* $\eta$ *is a natural homotopy equivalence.*

**Proof.** M associates each weakly directed poset to the set of its maximal elements. This mapping can be extended to a functor, as for each order-preserving map $f : P \to P'$, given a maximal element $l \in P$, there is a unique maximal element $l' \in P'$ with $f(l) \leq l'$. Given a set $S$, the maximal elements of the poset $(S, =)$ are all elements of $S$, so $\epsilon$ is a natural isomorphism. For a weak-directed poset $P$, the map $\eta_P : P \to \mathrm{Free}(\mathrm{M}(P))$ is a deformation retract of $P$ onto its core. To see this, starting from $P$, we can proceed by removing elements that are maximal in $P \setminus \mathrm{M}(P)$. If an elements is maximal in $P \setminus \mathrm{M}(P)$, it is necessarily upbeat: distinct maximal elements in $P$ can have no lower bound. After iteratively removing all elements in $P \setminus \mathrm{M}(P)$, we obtain the desired deformation retract $\eta_P : P \to \mathrm{Free}(\mathrm{M}(P))$. □

The functor M : **WDPoset**$_{\mathrm{m}}$ → **Set** induces a monic persistence function on **WDPoset**, by ([12], Prop. 3.6). Furthermore, such persistence function factors via a ranked category with finite colimits, **Set**, and is therefore stable by ([12], Thm. 3.27).

Universality is generally not granted for stable persistence functions. We now follow the logical line of Thm. 32 of [21] for proving the universality of the bottleneck (or matching) distance among the lower bounds for the natural pseudodistance that can come from distances between persistent block diagrams.

Let $F$ be a filtration in **WDPoset**. If $F(\infty)$ has several maximal elements, then all maximal elements arising in the filtration are bounded by one of them, so the construction can be performed for each of the lower set of maximal points of $F(\infty)$.

**Proposition 3.** *Let* $F, F'$ *be two filtrations in* **WDPoset***; let* $\mathcal{D}F$, $\mathcal{D}F'$ *be the respective persistence diagrams. Then there exist filtrations* $H, H'$ *such that*

1. $\mathcal{D}F = \mathcal{D}H,\ \mathcal{D}F' = \mathcal{D}H'$,
2. $d(\mathcal{D}H, \mathcal{D}H') = \delta(H, H')$,

*where d is the bottleneck distance between persistence diagrams. Therefore, d is universal with respect to the monic persistence function induced by* M.

**Proof.** There is at least one bijection $\gamma$ between the multisets $\mathcal{D}F$ and $\mathcal{D}F'$ which realizes the distance $d = d(\mathcal{D}F, \mathcal{D}F')$. Let $p_0$ and $p'_0$ be the multiplicities of the eldest cornerpoints at infinity of $\mathcal{D}F$, $\mathcal{D}F'$, respectively, and $p_1, \ldots, p_m, p'_1, \ldots, p'_m$ be the multiplicities of the remaining cornerpoints (possibly at infinity). We consider also cornerpoints on the diagonal, if they are needed to realize the matching distance. Up to relabeling the cornerpoints, we can assume that the matching is given by $p_i \mapsto p'_i$ for $i \in \{0, \ldots, m\}$. We denote $x_i, y_i$ (resp. $x'_i, y'_i$) the coordinates of the cornerpoint $p_i$ (resp. $p'_i$). The distance $d$ is then the maximum

of the distance in the $L^\infty$ norm of corresponding points. We now construct new filtrations of posets $H, H'$ as follows. See Figure 2 for a toy example.

$$H(x) = \{ p_i \mid x_i \leq x \}$$
$$p_i < p_j \text{ if } j = 0 \text{ and } y_i \leq x, \quad \text{and} \quad \begin{array}{l} H'(x) = \{ p'_i \mid x'_i \leq x \} \\ p'_i < p'_j \text{ if } j = 0 \text{ and } y'_i \leq x. \end{array}$$

Choosing the isomorphism $H(\infty) \to H'(\infty)$ given by $p_i \mapsto p'_i$, we can show that the pseudodistance between $H$ and $H'$ is smaller or equal than $d$. By stability, it must be equal. $\square$

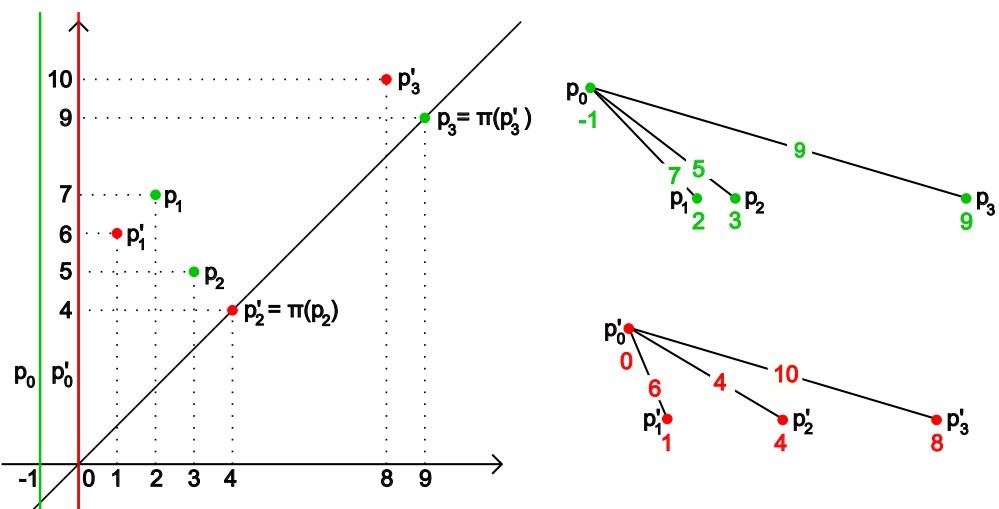

**Figure 2.** Two persistence diagrams $\mathcal{D}F$ (green), $\mathcal{D}F'$ (red) and the Hasse diagrams of the corresponding poset filtrations $H$, $H'$; edges are marked with the value at which the relation arises.

### 2.2. Weakly Directed Properties

Some of the most informative graph-theoretical concepts describe the local connectivity of a graph from different viewpoints, e.g., considering the number of edges to be removed to disconnect it. The following definitions express these kinds of stronger connectivities that we will use as categorical persistence functions, e.g., connected and biconnected components.

**Definition 7.** *By a property in a category **C** we mean a class $\mathcal{P} \subseteq \mathrm{Obj}(\mathbf{C})$ such that if an object $X$ belongs to $\mathcal{P}$ and object $Y$ is **C**-isomorphic to $X$, then also $Y$ belongs to $\mathcal{P}$. To say that $X$ has property $\mathcal{P}$ means that $X \in \mathcal{P}$.*

**Definition 8.** *Let **C** be a category and let $\mathcal{P}$ be a property. We call $S_\mathcal{P}$ the functor $\mathbf{C}_m \to \mathbf{Poset}_m$ that associates to each object in **C** the poset of its subobjects that have property $\mathcal{P}$. We say that the property $\mathcal{P}$ is weakly directed if, for all $X \in \mathrm{Obj}(\mathbf{C})$, $S_\mathcal{P}(X)$ is a weakly directed poset.*

**Proposition 4.** *Let **C** be a category, and let $\mathbf{C}_m$ be the subcategory of **C** where the only allowed morphisms are monomorphisms. Let $\mathcal{P}$ be a weakly directed property on $\mathrm{Obj}(\mathbf{C})$. Then $\mathcal{P}$ induces a stable categorical persistence function on $\mathbf{C}_m$, which we denote $p_\mathcal{P}$.*

**Proof.** We can consider the functor $S_\mathcal{P} \colon \mathbf{C}_m \to \mathbf{WDPoset}_m$. As $\mathbf{WDPoset}_m$ is equipped with a persistence function, this induces a persistence function on $\mathbf{C}_m$ by ([12], Prop. 3.3). $\square$

Unlike stability, the universality of $p_\mathcal{P}$ is in general not guaranteed. However, the following condition is sufficient to ensure it.

**Proposition 5.** *Let $\mathcal{P}, p_{\mathcal{P}}, S_{\mathcal{P}}$ be as in Definition 8 and Proposition 4. Let us further assume that there exists a functor $T\colon \mathbf{WDPoset}_m \to \mathbf{C}_m$ such that $\mathrm{M} \circ S_{\mathcal{P}} \circ T$ is naturally isomorphic to $\mathrm{M}$. That is to say, for all $Q \in \mathrm{Obj}(\mathbf{WDPoset}_m)$, the maximal elements of $S_{\mathcal{P}}(T(Q))$ are in a one-to-one correspondence with the maximal elements of $Q$, and this bijection is natural in P. Then the bottleneck distance between persistence diagrams is universal with respect to $p_{\mathcal{P}}$.*

**Proof.** Given two filtrations $F, F'$ in **C**, we can consider the filtrations of posets $S_{\mathcal{P}} \circ F$ and $S_{\mathcal{P}} \circ F'$. By Proposition 3, there are filtrations of weakly directed poset $H, H'$ with the same persistence diagram, whose interleaving distance equals the bottleneck distance. Then, $T \circ H$ and $T \circ H'$ have the same persistence diagram as $F, F'$, and their interleaving distance equals the bottleneck distance.   $\square$

## 3. Non-Simplicial Graph Persistence

In what follows, we fix $\mathbf{C} = \mathbf{Graph}$ and consequently $\mathbf{C}_m = \mathbf{Graph}_m$ is the subcategory of **Graph** in which only graph monomorphisms are allowed. We now translate some of the previous notions in terms of graphs.

**Definition 9.** *By a* property *we mean a set $\mathcal{P}$ of graphs such that if a graph X belongs to $\mathcal{P}$ and graph Y is isomorphic to X, then also Y belongs to $\mathcal{P}$. If $X \in \mathcal{P}$, we say that X has property $\mathcal{P}$.*

**Definition 10.** *Let $\mathcal{P}$ be a property. We call $S_{\mathcal{P}}$ the functor $\mathbf{Graph}_m \to \mathbf{Poset}_m$ that associates to each graph in **Graph** the poset of its subgraphs that have property $\mathcal{P}$. We say that the property $\mathcal{P}$ is* weakly directed *if, for all graphs X, $S_{\mathcal{P}}(X)$ is a weakly directed poset.*

**Remark 1.** *Topology is not really thrown out of the game. In fact, the nerve of the weakly directed poset $S_{\mathcal{P}}(X)$ is a simplicial complex. Let $\mathcal{D}_{\mathcal{P}}$ be the persistence diagram of an $(\mathbb{R}, \leq)$-indexed graph; it coincides with one of the persistence modules obtained by composing a chain of functors:*

$$(\mathbb{R}, \leq) \to \mathbf{Graph}_m \xrightarrow{S_{\mathcal{P}}} \mathbf{Poset}_m \xrightarrow{nerve} \mathbf{Simplicial\text{-}Sets} \xrightarrow{H_0} \mathbf{Vector\text{-}Spaces}$$

*Once more, we stress that our aim is to take the application-oriented researcher directly to the diagram, bypassing this detour.*

**Remark 2.** *The previous notions extend in a natural way to the case of $\mathbf{C} = \mathbf{Digraph}$, the category of directed graphs, discussed in Section 3.4.*

We now define a functor from $\mathbf{WDPoset}_m$ to $\mathbf{Graph}_m$, which will turn useful in the next subsections.

**Definition 11.** *Let n be a positive integer. Given a weakly directed poset Q, we can consider the graph whose vertices are $Q \times \{1, \ldots, n\}$, where distinct vertices $(v, i)$ and $(w, j)$ are connected by an edge if v and w are comparable in Q (see Figure 3 for an example of the constructions relative to the filtration H of Figure 2 and $n = 2, 3$). This mapping induces a functor $T_n\colon \mathbf{WDPoset}_m \to \mathbf{Graph}_m$.*

**Remark 3.** *The functor $T_n$ will be used to prove the universality of the bottleneck distance, with respect to persistence functions obtained from clique communities, k-connectedness, k-edge-connectedness, and strong connectedness. For each of those, one could find specific functors that produce simpler graphs, i.e., with fewer edges. However, we prefer to show a unified construction that works in a wide variety of cases.*

### 3.1. Clique Communities

An example of weakly directed property comes from clique communities. We recall the definition of clique community given in [24]. Given a graph $G = (V, E)$, two of its $k$-cliques (i.e., cliques of $k$ vertices) are said to be *adjacent* if they share $k - 1$ vertices; a $k-clique$

*community* is a maximal union of *k*-cliques such that any two of them are connected by a sequence of *k*-cliques, where each *k*-clique of the sequence is adjacent to the following one. This construction has been applied to network analysis [25–28] and to weighted graphs, in the classical topological persistence paradigm, in [3]. Here we consider a weighted graph as a filtration of graphs (where the weight of each vertex is the inf of the weights of its incident edges).

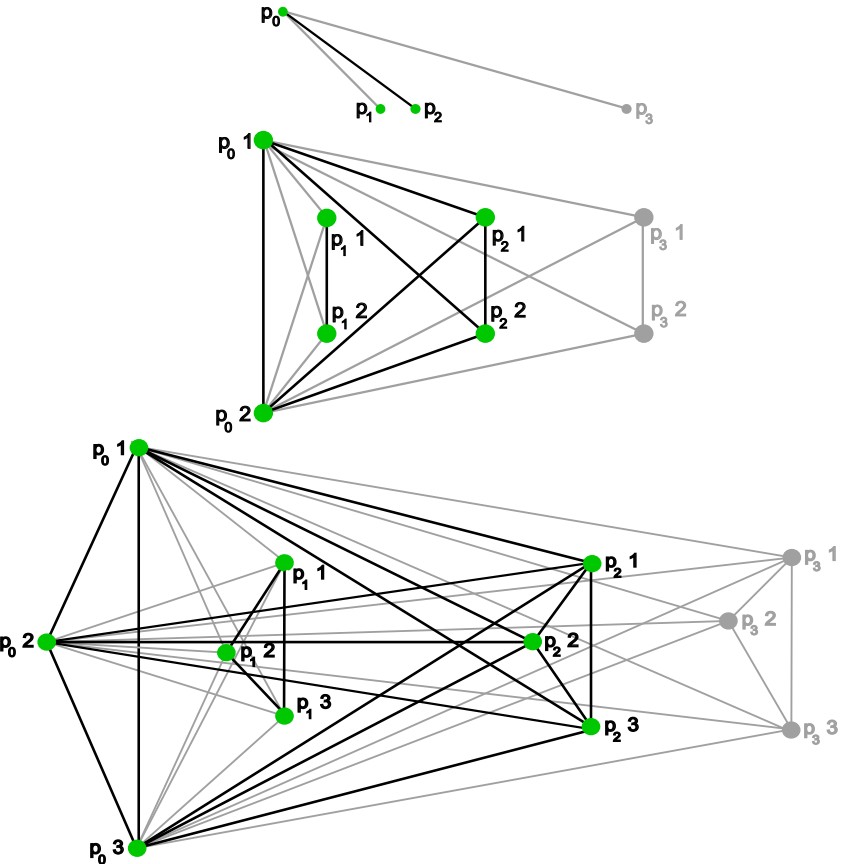

**Figure 3.** From the top: the inclusion of poset $H(5)$ into poset $H(\infty)$ of Figure 2, the image under $T_2$, and the image under $T_3$ (Definition 11).

**Definition 12.** *A graph G belongs to $c^k$ if it is union of k-cliques, such that any two of them are connected by a sequence of adjacent k-cliques.*

**Proposition 6.** *$c^k$ is a weakly directed property.*

**Proof.** If two subgraphs $G_1, G_2 \subseteq G$ are in $c^k$, and there is a *k*-clique in $G_1 \cap G_2$, then $G_1 \cup G_2$ is also in $c^k$. □

As a consequence, $c^k$ induces a stable persistence function $p_{c^k}$ on graph filtrations, which we call *persistent k-clique community number*. In practice, given a graph filtration *F*, the persistent *k*-clique community number $p_{c^k}(u, v)$ equals the number of *k*-clique communities in $F(v)$ that contain at least a *k*-clique when restricted to $F(u)$.

**Remark 4.** *Of course, the persistent 2-clique community number function of a weighted graph $(G, f)$, such that no isolated vertices appear in the filtration, coincides with its persistent 0-Betti number function.*

An example of persistent 3-clique community number function can be seen in Figure 4. We can associate to $p_{c^k}$, via ([12], Def. 3.13), a *persistent k-clique community diagram $D_{c^k}(f)$*.

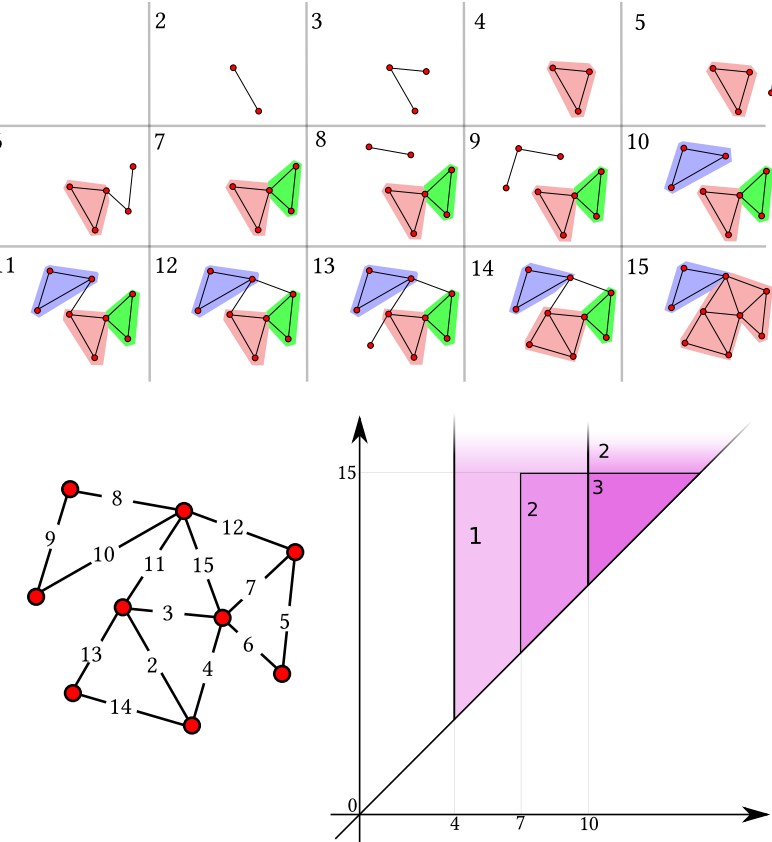

**Figure 4.** The weighted graph of Figure 1 (left), the corresponding filtration (above) and its 3-clique community number function.

**Proposition 7.** *The bottleneck distance is universal with respect to* $p_{c^k}$.

**Proof.** Let $T_k$ be the functor introduced in Definition 11. Then $M \circ S_{c^k} \circ T_k$ is naturally isomorphic to M. By Proposition 5, the bottleneck distance is universal with respect to $p_{c^k}$. $\square$

Figure 5 shows the two weighted graphs which realize the natural pseudodistance equal to the bottleneck distance between the persistence diagrams of Figure 2, when the persistence function is $p_{c^2}$. See the proof of Proposition 5 and the construction of Definition 11 for the underlying ideas.

*3.2. Blocks*

We recall that a connected graph is *k-vertex-connected* if it has at least $k$ vertices and remains connected whenever fewer than $k$ vertices are removed [29,30]. We say that a maximal $k$-vertex-connected subgraph of a given graph $G$ is a *k-vertex-connected component*.

Let us denote $v^k$ be the class of $k$-vertex-connected graphs. Since it is closed under isomorphisms, it is a property.

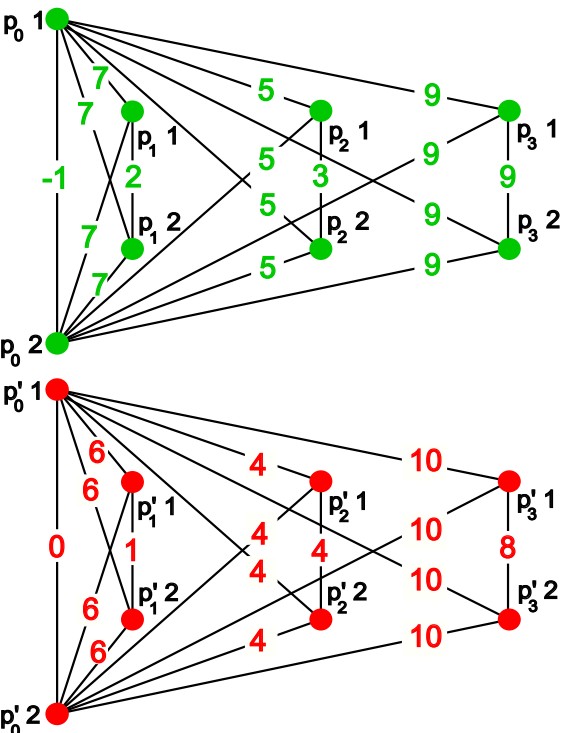

**Figure 5.** The weighted graphs whose natural pseudodistance equals the bottleneck distance of the persistence diagrams of Figure 2, relative to the persistence function $p_{c^2}$. Above (resp. below), the weighted graph corresponding to the green (resp. red) persistence diagram and to the upper (resp. lower) Hasse diagram of Figure 2.

**Proposition 8.** $v^k$ *is a weakly directed property.*

**Proof.** Let $G_1, G_2$ be $k$-vertex-connected subgraphs of a graph $G$, such that their intersection $G_3$ is $k$-vertex-connected. Let $U$ be any set of vertices of $G_1 \cup G_2$ with $|U| < k$. Then the induced subgraphs $G_1 - U, G_2 - U, G_3 - U$ are connected and the mutual intersections have at least one vertex. Since the union of connected graphs with nonempty intersection is connected, also $(G_1 \cup G_2) - U$ is connected. Therefore, $G_1 \cup G_2$ is $k$-vertex-connected. □

Property $v^k$ induces a functor

$$S_{v^k} : \mathbf{Graph}_{\mathrm{m}} \to \mathbf{WDPoset}_{\mathrm{m}}$$

and a stable monic persistence function $p_{v^k}$ on graph filtrations, which we call *persistent k-block number*. In practice, given a graph filtration $F$, the persistent $k$-block number $p_{c^k}(u, v)$ equals the number of $k$-vertex-connected components in $F(v)$ that contain at least a $k$-vertex-connected component when restricted to $F(u)$.

Furthermore, the bottleneck distance is universal with respect to the natural pseudodistance. To prove universality, we consider the same functor $T_n$ of Definition 11, and note that $M \circ S_{v^k} \circ T_k$ is naturally isomorphic to M.

An example of $k$-block number function (for $k = 2$) can be seen in Figure 6. We can then associate to $p_{v^k}$, via ([12], Def. 3.13), a *persistent block diagram* $D_{v^k}(f)$ with all classical features granted by the propositions of Section 2. A toy example is given in Figure 6.

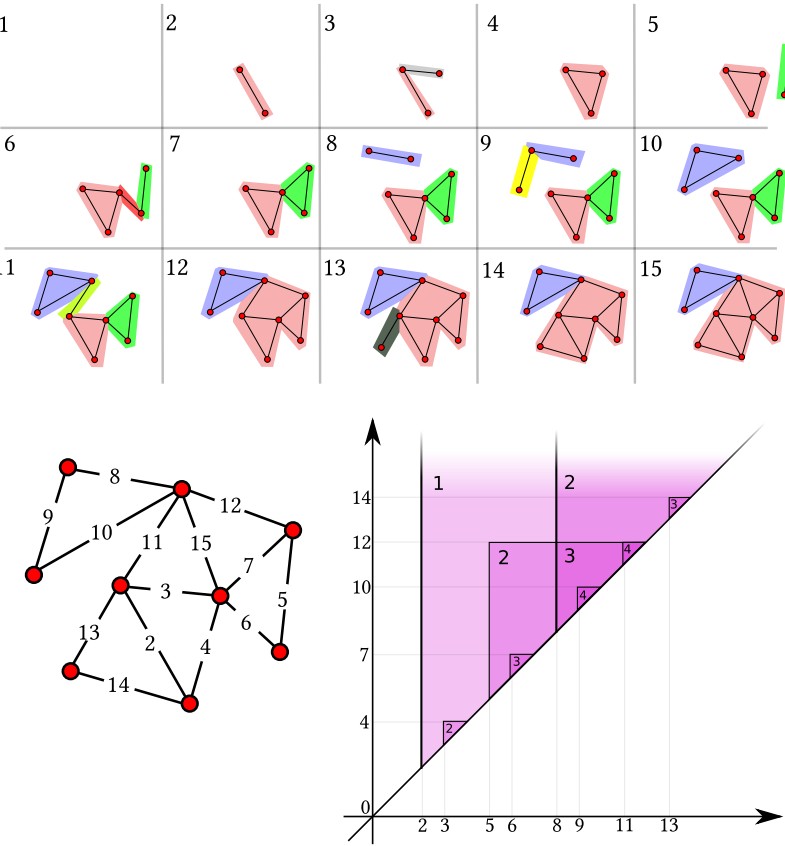

**Figure 6.** The weighted graph of Figure 1 (left), the corresponding filtration (above) and its 2-block number function.

### 3.3. Edge-Blocks

We say that a connected graph is *k-edge-connected* if it has at least *k* edges and remains connected whenever fewer than *k* edges are removed [30]. We say that a maximal *k*-edge-connected subgraph of a given graph *G* is a *k-edge-connected component*.

Let us denote $e^k$ be the class of *k*-edge-connected graphs. Since it is closed under isomorphisms, it is a property.

**Proposition 9.** $e^k$ *is a weakly directed property.*

**Proof.** Analogous to the proof of Proposition 8. □

Property $e^k$ therefore it induces a functor

$$S_{e^k} \colon \mathbf{Graph}_m \to \mathbf{WDPoset}_m$$

and a stable monic persistence function $p_{e^k}$ on graph filtrations, which we call *persistent k-edge-block number*. In practice, given a graph filtration *F*, the persistent *k*-edge-block number $p_{c^k}(u, v)$ equals the number of *k*-edge-connected components in $F(v)$ that contain at least a *k*-edge-connected component when restricted to $F(u)$.

Furthermore, the bottleneck distance is universal with respect to the natural pseudodistance. To prove universality, we consider the same functor $T_n$ of Definition 11, and note that $M \circ S_{e^k} \circ T_{k+1}$ is naturally isomorphic to M.

An example of persistent edge-block number function (for $k = 2$) can be seen in Figure 7. We can associate to $p_{e^k}$, via ([12], Def. 3.13), a *persistent edge-block diagram* $D_{e^k}(f)$.

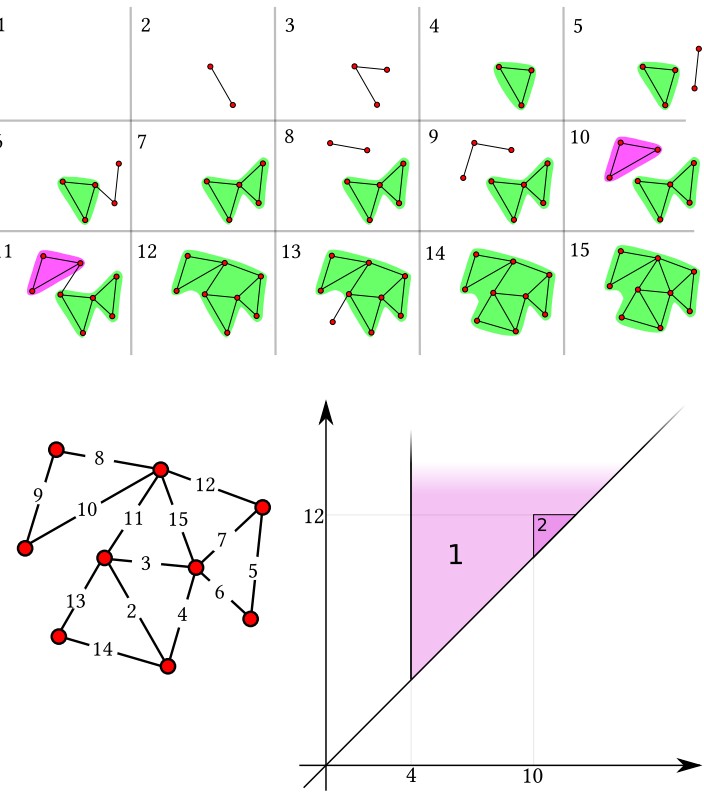

**Figure 7.** The weighted graph of Figure 1 (left), the corresponding filtration (above) and its 2-edge-block number function.

### 3.4. Strong Components in Digraphs

In this subsection, $\mathbf{C} = \mathbf{Digraph}$, the category of directed graphs and homomorphisms; $\mathbf{C_m} = \mathbf{Digraph}_m$ is its subcategory where only monomorphisms are allowed. A directed graph is *strongly connected* if for any pair of vertices $u, v$ there is a directed path from $u$ to $v$ (and one from $v$ to $u$) ([31], Section 3.4). A *strong component* of a digraph is a maximal strongly connected subdigraph.

We denote $s$ the class of strongly connected digraphs. Since it is closed under isomorphisms, it is a property.

**Proposition 10.** *$s$ is a weakly directed property.*

**Proof.** Immediate, since the strong components induce a partition of the vertex set. □

Hence, property $s$ induces a functor

$$S_s \colon \mathbf{Digraph}_m \to \mathbf{WDPoset}_m$$

and a stable monic persistence function $p_s$ on digraph filtrations, which we call *persistent strong component number*. In practice, given a digraph filtration $F$, the persistent strong component number $p_s(u, v)$ equals the number of strong components in $F(v)$ that contain at least a strong component when restricted to $F(u)$.

There exists a functor $\iota \colon \mathbf{Graph}_m \to \mathbf{Digraph}_m$, which replaces every undirected edge with a pair of directed edges with opposite orientations. The functor $\iota \circ T_n$ (see Definition 11), for any $n$, grants universality of the bottleneck distance. Figure 8 shows the persistent strong component diagram on two orientations of the usual graph, differing only on the edge with weight 2.

**Remark 5.** *The definitions of persistence functions of Sections 3.2 and 3.3 can easily be extended to digraphs.*

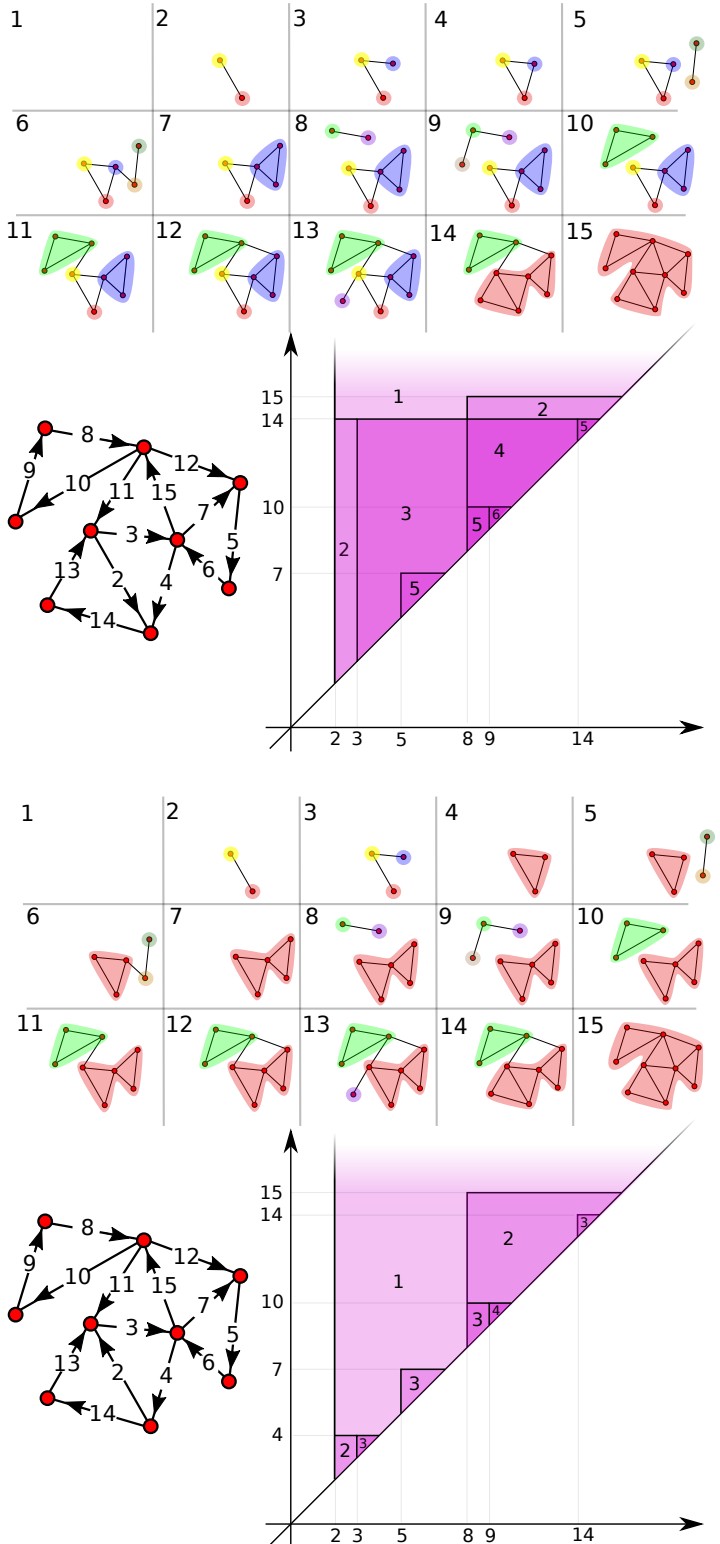

**Figure 8.** Two orientations of the weighted graph of Figure 1, differing by the orientation of the edge with weight 2. For each, the weighted digraph (left), the corresponding filtration (above) and its persistent strong component number function.

## 4. Conclusions and Perspectives

We built on a generalized theory of persistence, which no longer requires topological mediations such as auxiliary simplicial constructions, or the usage of homology as the functor of choice.

We gave a flexible definition—weakly directed property—for the construction of generalized persistence functions, and we applied them to toy examples in the category of weighted graphs. Therein, we discussed the stability and universality of the generalized persistence functions built following our definitions and considering blocks, edge-blocks, and clique communities.

This work is the combinatorial counterpart of the foundational results exposed in [12]. There, we focused on Abelian categories and categories of representations, aiming to extend the persistence to objects relevant in theoretical physics or theoretical chemistry: Lie-group representations, quiver representations, or representations of the category of cobordisms (related to topological quantum field theory in [32]). Here, we focus on categories that generally do not have any additive structure, but are of interest in several branches of Machine Learning and Artificial Intelligence. Graphs, digraphs, and their connectivity properties occupy a central role in these research fields, where networks are oftentimes represented as weighted digraphs. These structures can be compared quantitatively via stable categorical persistence functions, and possibly optimized by defining bottleneck-distance-based loss functions. We hope that this work paves the road for new applications of the persistence paradigm in various fields.

Finally, the posets considered here are homotopically discrete by Proposition 2. Properties leading to non homotopically discrete posets—with nontrivial higher homology groups—might give rise to new graph invariants.

**Author Contributions:** Conceptualization, M.G.B., M.F., P.V. and L. Z.; Writing —original draft, M.G.B., M.F., P.V.and L.Z.; Writing — review & editing, M.G.B., M.F., P.V. and L.Z. All authors have read and agreed to the published version of the manuscript.

**Funding:** Article written within the activity of INdAM-GNSAGA.

**Institutional Review Board Statement:** Not applicable.

**Informed Consent Statement:** Not applicable.

**Data Availability Statement:** Not applicable.

**Acknowledgments:** We are indebted to Diego Alberici, Emanuele Mingione, Pierluigi Contucci (whose research originated the present one), Luca Moci, Fabrizio Caselli, and Patrizio Frosini for many fruitful discussions. We thank the Reviewers for the useful suggestions and remarks.

**Conflicts of Interest:** The authors declare no conflict of interest.

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
