# Peer review of "Beyond Topological Persistence: Starting from Networks"

_mathematics, doi:10.3390/math9233079_

Round 1

Reviewer 1 Report

See the report. 

Author Response

We wish to heartfully thank Reviewer 1 for reading so carefully our paper and for giving us very enlightening suggestions.

Rev:
The authors stress and emphasize through out the entire article that what they present can not be obtained with topological constructions [...]

Answer:
We apologize for having appeared pretentious. We have reformulated some sentences in the Introduction. Please see the highlighted lines 15-21, 42-45.

Rev:
There are many ways to geometrize objects. [...]

Answer:
We definitely agree. We hope to have clarified our position by the addition of Remark 1 (highlighted lines 213-218), where we also took the liberty to copy the suggested functor composition.

Rev:
The authors also prove that the nerves of the considered posets are homotopically discrete. [...]

Answer:
Thank you very much for this interesting hint. We adopted it in the highlighted lines 348-350.

Reviewer 2 Report

This is a clearly written paper that describes particular examples of a category-theory type approach to persistence, but with the goal to generalise the “input domain” for persistence, rather than the “target domain”.   I found the paper very interesting and it is certain to be of broad appeal to researchers in both applied topology and in network analysis.  

I have one suggestion that the authors may like to follow up on.  There is related work defining a persistence module for strongly connected components in directed graphs in the following paper. 

K. Turner (2019)  “Rips filtrations for quasimetric spaces and asymmetric functions with stability results”  Algebraic and Geometric Topology, 19:1135–1170. 

This work is phrased using very different language, but I suspect at heart is making the same construction of persistence.  

Author Response

Thank you very much for your appreciation words!

Rev:
I have one suggestion that the authors may like to follow up on. [...]

Answer:
Thank you! We added the highlighted lines 21-23 in the Introduction and the related reference 11.
